# ASYMMETRIC SELF-PLAY FOR AUTOMATIC GOAL DISCOVERY IN ROBOTIC MANIPULATION

## ABSTRACT

We train a single, goal-conditioned policy that can solve many robotic manipulation tasks, including tasks with previously unseen goals and objects. We rely on asymmetric self-play for goal discovery, where two agents, Alice and Bob, play a game. Alice is asked to propose challenging goals and Bob aims to solve them. We show that this method can discover highly diverse and complex goals without any human priors. Bob can be trained with only sparse rewards, because the interaction between Alice and Bob results in a natural curriculum and Bob can learn from Alice's trajectory when relabeled as a goal-conditioned demonstration. Finally, our method scales, resulting in a single policy that can generalize to many unseen tasks such as setting a table, stacking blocks, and solving simple puzzles. Videos of a learned policy is available at `https://robotics-self-play.github.io`.

## 1 INTRODUCTION

We are motivated to train a *single* goal-conditioned policy (Kaelbling, 1993) that can solve *any* robotic manipulation task that a human may request in a given environment. In this work, we make progress towards this goal by solving a robotic manipulation problem in a table-top setting where the robot's task is to change the initial configuration of a variable number of objects on a table to match a given goal configuration. This problem is simple in its formulation but likely to challenge a wide variety of cognitive abilities of a robot as objects become diverse and goals become complex.

Motivated by the recent success of deep reinforcement learning for robotics (Levine et al., 2016; Gu et al., 2017; Hwangbo et al., 2019; OpenAI et al., 2019a), we tackle this problem using deep reinforcement learning on a very large training distribution. An open question in this approach is how we can build a training distribution rich enough to achieve generalization to many unseen manipulation tasks. This involves defining both an environment's initial state distribution and a goal distribution. The initial state distribution determines how we sample a set of objects and their configuration at the beginning of an episode, and the goal distribution defines how we sample target states given an initial state. In this work, we focus on a scalable way to define a rich goal distribution.

The research community has started to explore automated ways of defining goal distributions. For example, previous works have explored learning a generative model of goal distributions (Florensa et al., 2018; Nair et al., 2018b; Racaniere et al., 2020) and collecting teleoperated robot trajectories

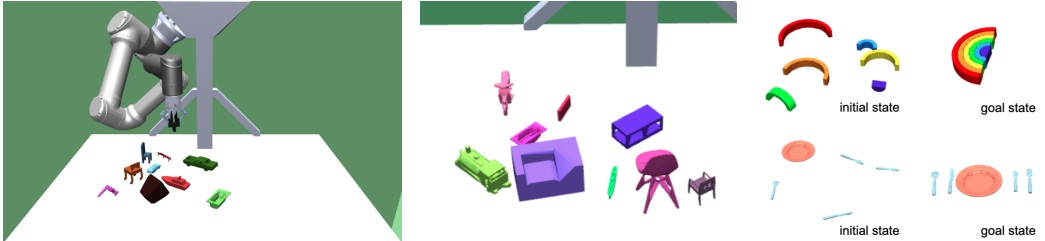

(a) Table-top setting with a robot arm    (b) Example initial state for training    (c) Example holdout tasks

Figure 1: (a) We train a policy that controls a robot arm operating in a table-top setting. (b) Randomly placed ShapeNet (Chang et al., 2015) objects constitute an initial state distribution for training. (c) We use multiple manually designed holdout tasks to evaluate the learned policy.

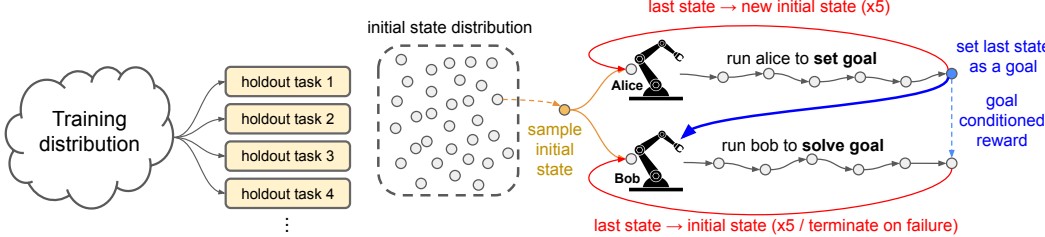

(a) Overall framework       (b) Training distribution based on asymmetric self-play

Figure 2: (a) We train a goal-conditioned policy on a single training distribution and evaluate its performance on many unseen holdout tasks. (b) To construct a training distribution, we sample an initial state from a predefined distribution, and run a goal setting policy (Alice) to generate a goal. In one episode, Alice is asked to generate 5 goals and Bob solves them in sequence until it fails.

to identify goals (Lynch et al., 2020; Gupta et al., 2020). In this paper, we extend an alternative approach called asymmetric self-play (Sukhbaatar et al., 2018b;a) for automated goal generation. Asymmetric self-play trains two RL agents named Alice and Bob. Alice learns to propose goals that Bob is likely to fail at, and Bob, a goal-conditioned policy, learns to solve the proposed goals. Alice proposes a goal by manipulating objects and Bob has to solve the goal starting from the same initial state as Alice's. By embodying these two agents into the same robotic hardware, this setup ensures that all proposed goals are provided with at least one solution: Alice's trajectory.

There are two main reasons why we consider asymmetric self-play to be a promising goal generation and learning method. First, any proposed goal is *achievable*, meaning that there exists at least one solution trajectory that Bob can follow to achieve the goal. Because of this property, we can exploit Alice's trajectory to provide additional learning signal to Bob via behavioral cloning. This additional learning signal alleviates the overhead of heuristically designing a curriculum or reward shaping for learning. Second, this approach does not require labor intensive data collection.

In this paper, we show that asymmetric self-play can be used to train a goal-conditioned policy for complex object manipulation tasks, and the learned policy can zero-shot generalize to many manually designed holdout tasks, which consist of either previously unseen goals, previously unseen objects, or both. To the best of our knowledge, this is the first work that presents zero-shot generalization to many previously unseen tasks by training purely with asymmetric self-play.[1]

## 2   Problem Formulation

Our training environment for robotic manipulation consists of a robot arm with a gripper attached and a wide range of objects placed on a table surface (Figure 1a, 1b). The goal-conditioned policy learns to control the robot to rearrange randomly placed objects (the initial state) into a specified goal configuration (Figure 1c). We aim to train a policy on a single training distribution and to evaluate its performance over a suite of holdout tasks which are independently designed and not explicitly present during training (Figure 2a). In this work, we construct the training distribution via *asymmetric self-play* (Figure 2b) to achieve generalization to many unseen holdout tasks (Figure 1c).

**Mathematical formulation** Formally, we model the interaction between an environment and a goal-conditioned policy as a goal-augmented Markov decision process $\mathcal{M} = \langle \mathcal{S}, \mathcal{A}, \mathcal{P}, \mathcal{R}, \mathcal{G} \rangle$, where $\mathcal{S}$ is the state space, $\mathcal{A}$ is the action space, $\mathcal{P} : \mathcal{S} \times \mathcal{A} \times \mathcal{S} \mapsto \mathbb{R}$ denotes the transition probability, $\mathcal{G} \subseteq \mathcal{S}$ specifies the goal space and $\mathcal{R} : \mathcal{S} \times \mathcal{G} \mapsto \mathbb{R}$ is a goal-specific reward function. A goal-augmented trajectory sequence is $\{(s_0, g, a_0, r_0), \ldots, (s_t, g, a_t, r_t)\}$, where the goal is provided to the policy as part of the observation at every step. We say a goal is achieved if $s_t$ is sufficiently close to $g$ (Appendix A.2). With a slightly overloaded notation, we define the *goal distribution* $\mathcal{G}(g|s_0)$ as the probability of a goal state $g \in \mathcal{G}$ conditioned on an initial state $s_0 \in \mathcal{S}$.

---

[1]Asymmetric self-play is proposed in Sukhbaatar et al. (2018b;a), but to supplement training while the majority of training is conducted on target tasks. Zero-shot generalization to unseen tasks was not evaluated.

**Training goal distribution** A naive design of the goal distribution $\mathcal{G}(g|s_0)$ is to randomly place objects uniformly on the table, but it is unlikely to generate interesting goals, such as an object picked up and held above the table surface by a robot gripper. Another possible approach, collecting tasks and goals manually, is expensive and hard to scale. We instead sidestep these issues and automatically generate goals via training based on asymmetric self-play (Sukhbaatar et al., 2018b;a). Asymmetric self-play involves using a policy named Alice $\pi_A(a|s)$ to set goals and a goal-conditioned policy Bob $\pi_B(a|s,g)$ to solve goals proposed by Alice, as illustrated in Figure 2b. We run $\pi_A$ to generate a trajectory $\tau_A = \{(s_0, a_0, r_0), \dots, (s_T, a_T, r_T)\}$ and the last state is labelled as a goal $g$ for $\pi_B$ to solve. The goal distribution $\mathcal{G}(s_T = g|s_0)$ is fully determined by $\pi_A$ and we train Bob only on this goal distribution. We therefore say *zero-shot generalization* when Bob generalizes to a holdout task which is not explicitly encoded into the training distribution.

**Evaluation on holdout tasks** To assess zero-shot generalization of $\pi_B(a|s,g)$ from our training setup, we hand-designed a suite of holdout tasks with goals that are never directly incorporated into the training distribution. Some holdout tasks also feature previously unseen objects. The holdout tasks are designed to either test whether a specific skill has been learned, such as the ability to pick up objects (Figure 3), or represent a semantically interesting task, such as setting a table (Figure 1c). Appendix B.6 describes the list of holdout tasks that we use in our experiments. Note that none of the holdout tasks are used for training $\pi_B(a|s,g)$.

## 3 ASYMMETRIC SELF-PLAY

To train Alice policy $\pi_A(a|s)$ and Bob policy $\pi_B(a|s,g)$, we run the following multi-goal game within one episode, as illustrated in Figure 2b:

1. An initial state $s_0$ is sampled from an initial state distribution. Alice and Bob are instantiated into their own copies of the environment. Alice and Bob alternate turns as follows.

2. **Alice's turn.** Alice interacts with its environment for a fixed number of $T$ steps and may rearrange the objects. The state at the end of Alice's turn $s_T$ will be used as a goal $g$ for Bob. If the proposed goal is invalid (e.g. if Alice has not moved any objects, or if an object has fallen off the table), the episode terminates.

3. **Bob's turn.** Bob receives reward if it successfully achieves the goal $g$ in its environment. Bob's turn ends when it succeeds at achieving the goal or reaches a timeout. If Bob's turn ends in a failure, its remaining turns are skipped and treated as failures, while we let Alice to keep generating goals.

4. Alice receives reward if Bob fails to solve the goal that Alice proposed. Steps 2–3 are repeated until 5 goals are set by Alice or Alice proposes an invalid goal, and then the episode terminates.

The competition created by this game encourages Alice to propose goals that are increasingly challenging to Bob, while Bob is forced to solve increasingly complex goals. The multi-goal setup was chosen to allow Bob to take advantage of environmental information discovered earlier in the episode to solve its remaining goals, which OpenAI et al. (2019a) found to be important for transfer to physical systems. Note however that in this work we focus on solving goals in simulation only. To improve stability and avoid forgetting, we have Alice and Bob play against past versions of their respective opponent in 20% of games. More details about the game structure and pseudocode for training with asymmetric self-play are available in Appendix A.

### 3.1 REWARD STRUCTURE

For Bob, we assign *sparse* goal-conditioned rewards. We measure the positional and rotational distance between an object and its goal state as the Euclidean distance and the Euler angle rotational distance, respectively. Whenever both distance metrics are below a small error (the *success threshold*), this object is deemed to be placed close enough to the goal state and Bob receives 1 reward immediately. But if this object is moved away from the goal state that it has arrived at in past steps, Bob obtains -1 reward such that the sum of per-object reward is at most 1 during a given turn. When all of the objects are in their goal state, Bob receives 5 additional reward and its turn is over.

For Alice, we assign a reward after Bob has attempted to solve the goal: 5 reward if Bob failed at solving the goal, and 0 if Bob succeeded. We shape Alice's reward slightly by adding 1 reward if it has set a valid goal, defined to be when no object has fallen off the table and any object has been moved more than the success threshold. An additional penalty of $-3$ reward is introduced when Alice sets a goal with objects outside of the placement area, defined to be a fixed 3D volume within the view of the robot's camera. More details are discussed in Appendix A.2.

### 3.2 ALICE BEHAVIORAL CLONING (ABC)

One of the main benefits of using asymmetric self-play is that the generated goals come with at least one solution to achieve it: *Alice's trajectory*. Similarly to Sukhbaatar et al. (2018a), we exploit this property by training Bob with Behavioral Cloning (BC) from Alice's trajectory, in addition to the reinforcement learning (RL) objective. We call this learning mechanism *Alice Behavioral Cloning* (ABC). We propose several improvements over the original formulation in Sukhbaatar et al. (2018a).

**Demonstration trajectory filtering** Compared to BC from expert demonstrations, using Alice's trajectory needs extra care. Alice's trajectory is likely to be suboptimal for solving the goal, as Alice might arrive at the final state merely by accident. Therefore, we only consider trajectories with goals that Bob failed to solve as demonstrations, to avoid distracting Bob with suboptimal examples. Whenever Bob fails, we relabel Alice's trajectory $\tau_A$ to be a goal-augmented version $\tau_{\text{BC}} = \{(s_0, s_T, a_0, r_0), \ldots, (s_T, s_T, a_T, r_T)\}$ as a demonstration for BC, where $s_T$ is the goal.

**PPO-style BC loss clipping** The objective for training Bob is $\mathcal{L} = \mathcal{L}_{\text{RL}} + \beta \mathcal{L}_{\text{abc}}$, where $\mathcal{L}_{\text{RL}}$ is an RL objective and $\mathcal{L}_{\text{abc}}$ is the ABC loss. $\beta$ is a hyperparameter controlling the relative importance of the BC loss. We set $\beta = 0.5$ throughout the whole experiment. A naive BC loss is to minimize the negative log-likelihood of demonstrated actions, $-\mathbb{E}_{(s_t, g_t, a_t) \in \mathcal{D}_{\text{BC}}} \left[ \log \pi_B(a_t | s_t, g_t; \theta) \right]$ where $\mathcal{D}_{\text{BC}}$ is a mini-batch of demonstration data and $\pi_B$ is parameterized by $\theta$. We found that overly-aggressive policy changes triggered by BC sometimes led to learning instabilities. To prevent the policy from changing too drastically, we introduce PPO-style loss clipping (Schulman et al., 2017) on the BC loss by setting the advantage $\hat{A} = 1$ in the clipped surrogate objective:

$$\mathcal{L}_{\text{abc}} = -\mathbb{E}_{(s_t, g_t, a_t) \in \mathcal{D}_{\text{BC}}} \left[ \text{clip}\left( \frac{\pi_B(a_t | s_t, g_t; \theta)}{\pi_B(a_t | s_t, g_t; \theta_{\text{old}})}, 1 - \epsilon, 1 + \epsilon \right) \right]$$

where $\pi_B(a_t | s_t, g_t; \theta)$ is Bob's likelihood on a demonstration based on the parameters that we are optimizing, and $\pi_B(a_t | s_t, g_t; \theta_{\text{old}})$ is the likelihood based on Bob's behavior policy (at the time of demonstration collection) evaluated on a demonstration. This behavior policy is identical to the policy that we use to collect RL trajectories. By setting $\hat{A} = 1$, this objective optimizes the naive BC loss, but clips the loss whenever $\frac{\pi_B(a_t | s_t, g_t; \theta)}{\pi_B(a_t | s_t, g_t; \theta_{\text{old}})}$ is bigger than $1 + \epsilon$, to prevent the policy from changing too much. $\epsilon$ is a clipping threshold and we use $\epsilon = 0.2$ in all the experiments.

## 4 RELATED WORK

**Training distribution for RL** In the context of multi-task RL (Beattie et al., 2016; Hausman et al., 2018; Yu et al., 2020), multi-goal RL (Kaelbling, 1993; Andrychowicz et al., 2017), and meta RL (Wang et al., 2016; Duan et al., 2016), previous works manually designed a distribution of tasks or goals to see better generalization of a policy to a new task or goal. Domain randomization (Sadeghi & Levine, 2017b; Tobin et al., 2017; OpenAI et al., 2020) manually defines a distribution of simulated environments, but in service of generalizing to the same task in the real world.

There are approaches to grow the training distribution automatically (Srivastava et al., 2013). Self-play (Tesauro, 1995; Silver et al., 2016; 2017; Bansal et al., 2018; OpenAI et al., 2019b; Vinyals et al., 2019) constructs an ever-growing training distribution where multiple agents learn by competing with each other, so that the resulting agent shows strong performance on a single game. OpenAI et al. (2019a) automatically grew a distribution of domain randomization parameters to accomplish better generalization in the task of solving a Rubik's cube on the physical robot. Wang et al. (2019;

2020) studied an automated way to keep discovering challenging 2D terrains and locomotion policies that can solve them in a 2D bipedal walking environment.

We employ asymmetric self-play to construct a training distribution for learning a goal-conditioned policy and to achieve generalization to unseen tasks. Florensa et al. (2018); Nair et al. (2018b); Racaniere et al. (2020) had the same motivation as ours, but trained a generative model instead of a goal setting policy. Thus, the difficulties of training a generative model were inherited by these methods: difficulty of modeling a high dimensional space and generation of unrealistic samples. Lynch et al. (2020); Gupta et al. (2020) used teleoperation to collect arbitrary robot trajectories, and defined a goal distribution from the states in the collected trajectories. This approach likely requires a large number of robot trajectories for each environment configuration (e.g. various types of objects on a table), and randomization of objects was not studied in this context.

**Asymmetric self-play** Asymmetric self-play was proposed by Sukhbaatar et al. (2018b) as a way to supplement RL training. Sukhbaatar et al. (2018b) mixed asymmetric self-play training with standard RL training on the target task and measured the performance on the target task. Sukhbaatar et al. (2018a) used asymmetric self-play to pre-train a hierarchical policy and evaluated the policy after fine-tuning it on a target task. Liu et al. (2019) adopted self-play to encourage efficient learning with sparse reward in the context of an exploration competition between a pair of agents. As far as we know, no previous work has trained a goal-conditioned policy *purely* based on asymmetric self-play and evaluated generalization to unseen holdout tasks.

**Curriculum learning** Many previous works showed the difficulty of RL and proposed an automated curriculum (Andrychowicz et al., 2017; Florensa et al., 2017; Salimans & Chen, 2018; Matiisen et al., 2019; Zhang et al., 2020) or auxiliary exploration objectives (Oudeyer et al., 2007; Baranes & Oudeyer, 2013; Pathak et al., 2017; Burda et al., 2019; Ecoffet et al., 2019; 2020) to learn *predefined tasks*. When training goal-conditioned policies, relabeling or reversing trajectories (Andrychowicz et al., 2017; Florensa et al., 2017; Salimans & Chen, 2018) or imitating successful demonstrations (Oh et al., 2018; Ecoffet et al., 2019; 2020) naturally reduces the task complexity. Our work shares a similarity in that asymmetric self-play alleviates the difficulty of learning a goal-conditioned policy via an intrinsic curriculum and imitation from the goal setter's trajectory, but our work does not assume any predefined task or goal distribution.

**Hierarchical reinforcement learning (HRL)** Some HRL methods jointly trained a goal setting policy (high-level or manager policy) and a goal solving policy (low-level or worker policy) (Vezhnevets et al., 2017; Levy et al., 2019; Nachum et al., 2018). However, the motivation for learning a goal setting policy in HRL is not to challenge the goal solving policy, but to cooperate to tackle a task that can be decomposed into a sequence of sub-goals. Hence, this goal setting policy is trained to optimize task reward for the target task, unlike asymmetric self-play where the goal setter is rewarded upon the other agent's failure.

**Robot learning for object manipulation.** It has been reported that training a policy for multi-object manipulation is very challenging with *sparse* rewards (Riedmiller et al., 2018; Vecerik et al., 2018). One example is block stacking, which has been studied for a long time in robotics as it involves complex contact reasoning and long horizon motion planning (Deisenroth et al., 2011). Learning block stacking often requires a hand-designed curriculum (Li et al., 2019), meticulous reward shaping (Popov et al., 2017), fine-tuning (Rusu et al., 2017), or human demonstrations (Nair et al., 2018a; Duan et al., 2017). In this work, we use block stacking as one of the holdout tasks to test zero-shot generalization, but without training on it.

## 5 EXPERIMENTS

In this section, we first show that asymmetric self-play generates an effective training curriculum that enables generalization to unseen hold-out tasks. Then, the experiment is scaled up to train in an environment containing multiple random complex objects and evaluate it with a set of holdout tasks containing unseen objects and unseen goal configurations. Finally, we demonstrate how critical ABC is for Bob to make progress in a set of ablation studies.

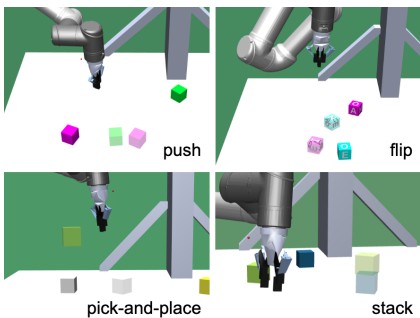

Figure 3: Holdout tasks in the environment using 1 or 2 blocks. The transparent blocks denote the desired goal state, while opaque blocks are the current state. (a) `push`: The blocks must be moved to their goal locations and orientations. There is no differentiation between the six block faces. (b) `flip`: Each side of the block is labelled with a unique letter. The blocks must be moved to make every face correctly positioned as what the goal specifies. (c) `pick-and-place`: One goal block is in the air. (d) `stack`: Two blocks must be stacked in the right order at the right location.

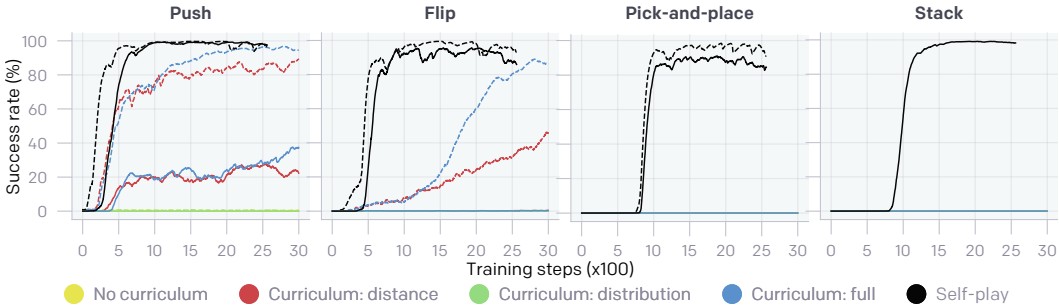

Figure 4: Generalization to unseen holdout tasks for blocks. Baselines are trained over a mixture of all holdout tasks. The solid lines represent 2-blocks, while the dashed lines are for 1-block. The x-axis denotes the number of training steps via asymmetric self-play. The y-axis is the zero-shot generalization performance of Bob policy at corresponding training checkpoints. Note that success rate curves of completely failed baselines are occluded by others.

## 5.1 EXPERIMENTAL SETUP

We implement the training environment[2] described in Sec. 2 with randomly placed ShapeNet objects (Chang et al., 2015) as an initial state distribution. In addition, we set up another simpler environment using one or two blocks of fixed size, used for small-scale comparisons and ablation studies. Figure 3 visualizes four holdout tasks for this environment. Each task is designed to evaluate whether the robot has acquired certain manipulation skills: pushing, flipping, picking up and stacking blocks. Experiments in Sec. 5.2, 5.3 and 5.5 focus on blocks and experimental results based on ShapeNet objects are present on Sec. 5.4. More details on our training setups are in Appendix B.

We implement Alice and Bob as two independent policies of the same network architecture with memory (Appendix B.4), except that Alice has no observation on goal state. The policies take state observations ("state policy") for experiments with blocks (Sec. 5.2, 5.3, and 5.5), and take both vision and state observations ("hybrid policy") for experiments with ShapeNet objects (Sec. 5.4). Both policies are trained with Proximal Policy Optimization (PPO) (Schulman et al., 2017).

## 5.2 GENERALIZATION TO UNSEEN GOALS WITHOUT MANUAL CURRICULA

One way to train a single policy to acquire all the skills in Figure 3 is to train a goal-conditioned policy directly over a mixture of these tasks. However, training directly over these tasks without a curriculum turns out to be very challenging, as the policy completely fails to make any progress.[3] In contrast, Bob is able to solve all these holdout tasks quickly when learning via asymmetric self-play, without explicitly encoding any prior knowledge of the holdout tasks into the training distribution.

To gauge the effect of an intrinsic curriculum introduced by self-play, we carefully designed a set of non-self-play baselines using explicit curricula controlled by Automatic Domain Randomization (OpenAI et al., 2019a). All baselines are trained over a mixture of block holdout tasks as the

---

[2]Our training and evaluation environments are publicly available at `hide-for-anonymous-purpose`

[3]The tasks was easier when we ignored object rotation as part of the goal, and used a smaller table.

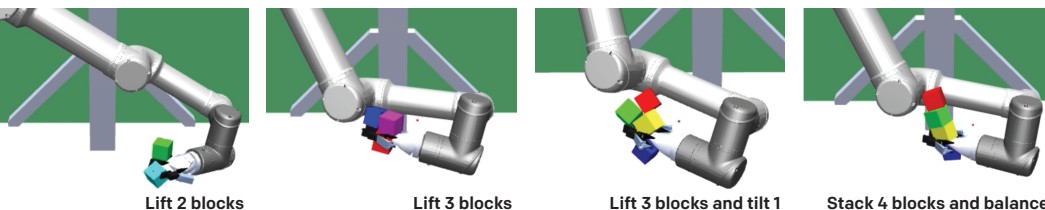

**Lift 2 blocks**     **Lift 3 blocks**     **Lift 3 blocks and tilt 1**     **Stack 4 blocks and balance**

Figure 5: Goals discovered by asymmetric self-play. Alice discovers many goals that are not covered by our manually designed holdout tasks on blocks.

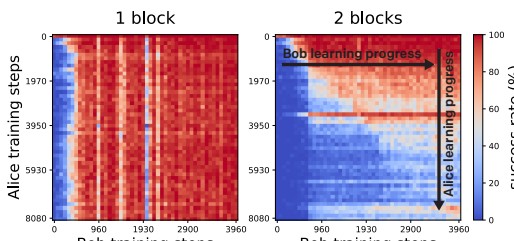

Figure 6: The empirical payoff matrix between Alice and Bob. Average success rate over multiple self-play episode is visualized. Alice with more training steps generates more challenging goals that Bob cannot solve yet. Bob with more training steps can achieve more goals against the same Alice.

goal distribution. We measure the effectiveness of a training setup by tracking the success rate for each holdout task, as shown in Figure 4. The `no curriculum` baseline fails drastically. The `curriculum:distance` baseline expands the distance between the initial and goal states gradually as training progresses, but only learns to push and flip a single block. The `curriculum:distribution` baseline, which slowly increases the proportion of pick-and-place and stacking goals in the training distribution, fails to acquire any skill. The `curriculum:full` baseline incorporates all hand-designed curricula yet still cannot learn how to pick up or stack blocks. We have spent a decent amount of time iterating and improving these baselines but found it especially difficult to develop a scheme good enough to compete with asymmetric self-play. See Appendix C.1 for more details of our baselines.

## 5.3 DISCOVERY OF NOVEL GOALS AND SOLUTIONS

Asymmetric self-play discovers novel goals and solutions that are not covered by our holdout tasks. As illustrated in Figure 5, Alice can lift multiple blocks at the same time, build a tower and then keep it balanced using an arm joint. Although it is a tricky strategy for Bob to learn on its own, with ABC, Bob eventually acquires the skills for solving such complex tasks proposed by Alice. Videos are available at `https://robotics-self-play.github.io`.

Figure 6 summarizes Alice and Bob's learning progress against each other. For every pair of Alice and Bob, we ran multiple self-play episodes and measured the success rate. We observe an interesting trend with 2 blocks. As training proceeds, Alice tends to generate more challenging goals, where Bob shows lower success rate. With past sampling, Bob continues to make progress against versions of Alices from earlier optimization steps. This visualization suggests a desired dynamic of asymmetric self-play that could potentially lead to unbounded complexity: Alice continuously generates goals to challenge Bob, and Bob keeps making progress on learning to solve new goals.

## 5.4 GENERALIZATION TO UNSEEN OBJECTS AND GOALS

The experiments above show strong evidence that efficient curricula and novel goals can autonomously emerge in asymmetric self-play. To further challenge our approach, we scale it up to work with many more complex objects using more computational resources for training. We train a hybrid policy in an environment containing up to 10 random ShapeNet (Chang et al., 2015) objects. During training, we randomize the number of objects and the object sizes via Automatic Domain Randomization (OpenAI et al., 2019a). The hybrid policy uses vision observations to extract information about object geometry and size. We evaluate the Bob policy on a more diverse set of manipulation tasks, including semantically interesting ones. Many tasks contain unseen objects and complex goals, as illustrated in Figure 7.

The learned Bob policy achieves decent zero-shot generalization performance for many tasks. Success rates are reported in Figure 8. Several tasks are still challenging. For example, `ball-capture`

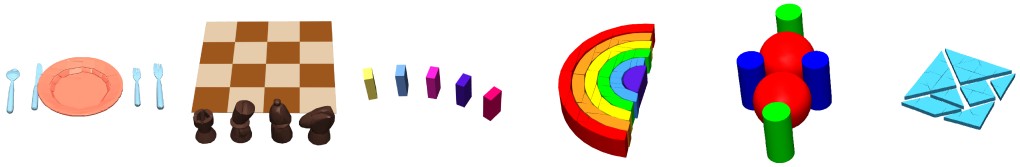

(a) Table setting   (b) Mini chess   (c) Domino   (d) Rainbow   (e) Ball-capture   (f) Tangram

Figure 7: Example holdout tasks involving unseen objects and complex goal states. The goal states are illustrated here, and the initial states have randomly placed objects.

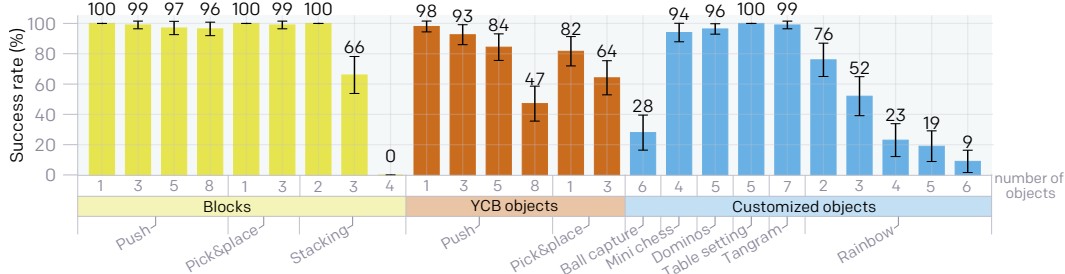

Figure 8: Success rates of a single goal-conditioned policy solving a variety of holdout tasks, averaged over 100 trials. The error bars indicate the 99% confidence intervals. Yellow, orange and blue bars correspond to success rates of manipulation tasks with blocks, YCB[4] objects and other uniquely built objects, respectively. Videos are available at `https://robotics-self-play.github.io`.

requires delicate handling of rolling objects and lifting skills. The `rainbow` tasks call for an understanding of concave shapes. Understanding the ordering of placement actions is crucial for stacking more than 3 blocks in the desired order. The Bob policy learns such an ordering to some degree, but fails to fully generalize to an arbitrary number of stacked blocks.

## 5.5 ABLATION STUDIES

We present a series of ablation studies designed for measuring the importance of each component in our asymmetric self-play framework, including Alice behavioral cloning (ABC), BC loss clipping, demonstration filtering, and the multi-goal game setup. We disable a single ingredient in each ablation run and compare with the complete self-play baseline in Figure 9.

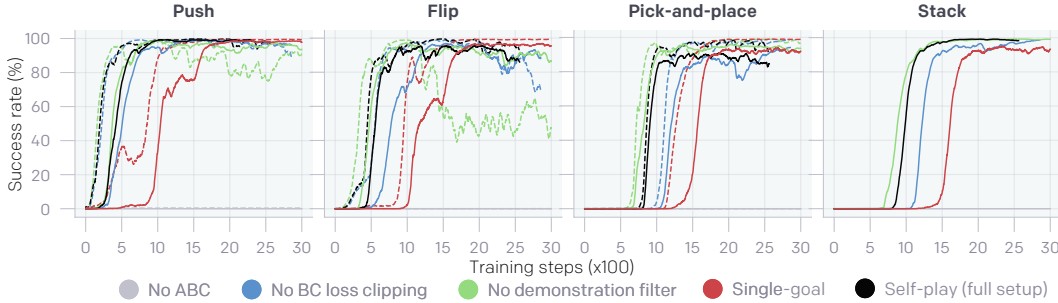

Figure 9: The ablation studies compare four ablation runs each with one component disabled with the full baseline. Solid lines are for 2-blocks, dashed lines are for 1-block. The x-axis denotes the number of training steps via asymmetric self-play. The y-axis is the zero-shot generalization performance of Bob policy at corresponding training steps.

The `no ABC` baseline shows that Bob *completely fails* to solve any holdout task without ABC, indicating that ABC is a critical mechanism in asymmetric self-play. The `no BC loss clipping` baseline shows slightly slower learning on `pick-and-place` and `stack`, as well as some instabilities in the middle of training. Clipping in the BC loss is expected to help alleviate this instability by con-

trolling the rate of policy change per optimizer iteration. The `no demonstration filter` baseline shows noticeable instability on `flip`, suggesting the importance of excluding suboptimal demonstrations from behavioral cloning. Finally, the `single-goal` baseline uses a single goal instead of 5 goals per episode during training. The evaluation tasks are also updated to require a single success per episode. Generalization of this baseline to holdout tasks turns out to be much slower and less stable. It signifies some advantages of using multiple goals per episode, perhaps due to the policy memory internalizing environmental information during multiple trials of goal solving.

The results of the ablation studies suggest that ABC with proper configuration and multi-goal gameplay are critical components of asymmetric self-play, alleviating the importance of manual curricula and facilitating efficient learning.

## 6  CONCLUSION

One limitation of our asymmetric self-play approach is that it depends on a resettable simulation environment as Bob needs to start from the same initial state as Alice's. Therefore asymmetric self-play training has to happen in a simulator which can be easily updated to a desired state. In order to run the goal-solving policy on physical robots, we plan to adopt sim-to-real techniques in future work. Sim-to-real has been shown to achieve great performance on many robotic tasks in the real world (Sadeghi & Levine, 2017a; Tobin et al., 2017; James et al., 2019; OpenAI et al., 2020). One potential approach is to pre-train two agents via asymmetric self-play in simulation, and then fine-tune the Bob policy with domain randomization or data collected on physical robots.

In conclusion, we studied asymmetric self-play as a framework for defining a single training distribution to learn many arbitrary object manipulation tasks. Even without any prior knowledge about the target tasks, asymmetric self-play is able to train a strong goal-conditioned policy that can generalize to many unseen holdout tasks. We found that asymmetric self-play not only generates a wide range of interesting goals but also alleviates the necessity of designing manual curricula for learning such goals. We provided evidence that using the goal setting trajectory as a demonstration for training a goal solving policy is essential to enable efficient learning. We further scaled up our approach to work with various complex objects using more computation, and achieved zero-shot generalization to a collection of challenging manipulation tasks involving unseen objects and unseen goals.

---

[4]https://www.ycbbenchmarks.com/object-models/

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
