# OpenReview forum: "Asymmetric self-play for automatic goal discovery in robotic manipulation"
_ICLR.cc/2021/Conference — Reject_

### Official Review · AnonReviewer4 · 2020-10-28
**slight variation on asymmetric self-play extended to robotics tasks (only in sim) and evaluation on hold-out tasks**

**Rating:** 6
**Confidence:** 3

**Review:**

The paper proposes a variation of asymmetric self play (Sukhbaatar et al. 2018b) where:
- Bob gets to use Alice's trajectories for Behaviour Cloning (ABC) training (with some logic around when best to employ this).
- There may be several goals (reward-related) per episode and extra care is taken to avoid training on bad episodes (e.g. items didn't move or items fell off table)
- Tasks make use of a simulated robotic arm environment (however the observations include ground truth data that make the sim far from anything real)
- Some study-specific details of training methods (PPO-like) and agent network architectures (in suppl) which may be of interest.

At first glance, the movies that accompany this paper show a surprisingly high level of performance for the given task complexity.  This surprise was reduces when by the supplementary material where it is clearly explained that the observations include the full ground-truth information on the robot state and the manipulated objects: "object state observation contains each object’s position, rotation, velocity, rotational velocity, the distance between the object and the gripper, as well as whether this object has contacts with the gripper." In my view, this renders the experiments deep within the realm of toy simulations. I also feel that showing movies of vision inputs is a bit misleading because, in my view, it coveys a message that this work is on vision-based manipulation learning while in reality there is nearly full knowledge of all the items' true states.

There are multiple figures showing the merit of the approach within the designed task framework in the form of comparison with baseline methods that do not use the asymmetric-self play framework and an ablation study. ~~The one comparison with (Sukhbaatar et al. 2018b), which this study is a variation of, appears at the very end of the supplementary material and seems to show that the two methods do equally well on tasks presented in this study. This last point leads me to conclude that the variations from (Sukhbaatar et al. 2018b) in this study do not harm but also do not improve on the original asymmetric-self-play model.~~

**EDIT: per the explanation in the author feedback (not a direct comparison with ukhbaatar et al. 2018b), that conclusion is unwarranted.**

I found the paper lacking in clarity regarding both method and evaluation:
The method:
Section 3 describes the method. It reads more like a summary pointing out the important bits than an ordered description that would allow someone to replicate the framework and model updates (which seem to all be tightly coupled in that the running of Alice and Bob, the generation of rewards and the rules around when to do what type of model update step all contribute to the framework whose product is two agents and the tasks that they set up for each-other).
~~If the process could be described as a set of agents and interfaces and pseudocode, perhaps the reader would get a clearer understanding of how they might replicate the experiments. Note also that code is not provided.~~

**EDIT: pseudocode (and potential future link to code) added**


Evaluation:
It is clear that evaluation was performed on a large number of tasks and task variations and there is information regarding the tasks and models and parameters in the supplementary material. What I found to be unclear is - how were the experiments on hold-out tasks different from the other tasks. More specifically, in what ways is the framework used differently during hold out tasks. Is asymmetric self play used? Is Alice Behavioural Cloning applied?  If so, then in what sense are these tasks held out? If not, then how is the Bob agent trained for these tasks? The unclarity in the description of the method may have contributed to the unclarity of the evaluation section (in that it is not clear to me how the framework is modified between these task groups).

**EDIT: Thanks to clarifications in author feedback and some modifications to the manuscript I find the paper clearer w.r.t. evaluation (though still feel that reading the paper only may leave the reader confused).**

The reliance on behavioural cloning (shown to be crucial to achieving non-zero success in this study's modelling) limits the proposed method to cases where the environment can be reset which would make it much harder to implement outside of a simulation environment.

Overall, due to the issues mentioned above, I find this study to add little to the methodology and understanding of asymmetric self play as an RL method for robotic manipulation.

**EDIT: with author feedback and changes to manuscript (and supplementary) I think that the study is more interesting than expressed in my original review, however the sim-real gap with respect to applicability to robotics is still a major concern IMO. I have updated my score accordingly.**

---

> ### Author Response · Authors · 2020-11-13
> **Thank you for the detailed review! Please check the discussion regarding some confusion and we have updated the paper accordingly.**
>
> Thank you for your reviews! Based on the feedback, we have done the following updates in a new version of the manuscript:
> 1. We added clarification in captions of Figure 4 & 9 to describe how we evaluate the goal-conditioned policy in more detail, to avoid confusion.
> 2. We added pseudocode for asymmetric self-play training episodes in Appendix A.4.
> 3. We added discussion regarding the limitations of our work in Sec. 6.
> 4. We will open-source our simulation environments for both training and evaluation. We added the link in the paper but it is currently masked for anonymous reason.
>
> We truly appreciate your time and detailed comments. We have responded to some review points as follows.
>
> **1. Main contribution: zero-shot generalization using asymmetric self-play**
> Our main contribution is using asymmetric self-play for zero-shot generalization to many unseen holdout tasks without using these tasks for training. The biggest difference between our work and [Sukhbaatar et al. 2018b](https://arxiv.org/abs/1703.05407) is that we studied asymmetric self-play in zero-shot generalization setup, and showed evidence that asymmetric self-play, without any prior on the goal distribution, can achieve good performance. On the contrary, all the experiments in Sukhbaatar et al. 2018b mixes asymmetric self-play with a standard RL training on target task, so zero-shot performance cannot be evaluated. We refer the reviewer to Appendix B of Sukhbaatar et al. 2018b where they clarified that they used asymmetric self-play for **1-20% of the training**, so the majority of training data still comes from the same task distribution for evaluation. In contrast, we train both Alice and Bob policies **100% with self-play** and we do not use any holdout tasks for training,
>
> We greatly appreciated previous work by Sukhbaatar et al. 2018b and enjoyed their work. However, training a policy purely with asymmetric self-play for solving robotic manipulation with an unbounded set of objects and goals is an idea that was not proposed or studied in the previous work. The possibility of fully relying on self-play to create *an ever-growing complex distribution of goals* that can lead to zero-shot generalization, as shown in our paper, truly excites us and we believe it is worthy of sharing with the community.
>
> **2. Training / evaluation setup: holdout tasks are not used for training**
> In this work, holdout tasks are not used for training at all. For example, in the block tasks, we use a training environment with 1-2 blocks without any predefined goals. We use asymmetric self-play with ABC in this environment to train a goal-conditioned policy, Bob. At test time, we run Bob policy in holdout tasks with goals corresponding to different manipulation skills (push, flip, pick-and-place, and stacking). For Bob, this is zero-shot because we didn’t explicitly include any of those manipulation goals into the training data. Alice is not involved at test time. Neither asymmetric self-play nor ABC is used for evaluation on holdout tasks.
>
> To describe the details of how we conduct experiments, we will describe how we generate plots in Figure 4 & Figure 9 as an example. We use a training environment containing 1 or 2 blocks **without any pre-defined goal**. Alice and Bob are trained in this environment with asymmetric self-play and ABC. Note that Bob is a goal-conditioned policy, but we do not provide a goal and instead Alice produces a goal for Bob. During training, we checkpoint goal-conditioned policy (Bob) for every training step.
> At test time, we take every checkpointed version of the goal-conditioned policy and evaluate it on the holdout tasks (push, flip, pick-and-place, and stack) where the goal is pre-defined. We only measure the success rate of Bob’s rollouts in holdout tasks. No training or gradient updates of policy weights are involved at test time. When we draw plots in Figure 4 and Figure 9, x-axis denotes the number of asymmetric self-play training steps for each checkpoint and y-axis is the zero-shot performance of Bob policy.
>
> We appreciate your comment on your confusion for the evaluation setting. We have improved the manuscript by clarifying the commented points.
>
> (to be continued)

---

> > ### Author Response · Authors · 2020-11-13
> > **(cont'd)**
> >
> > **3. Comparisons with Sukhbaatar et al. 2018b**
> > The comaprison with timestep-based reward in Appendix C.2 is not the comparison with [Sukhbaatar et al. 2018b](https://arxiv.org/abs/1712.00948). It is an ablation study comparing two different reward functions based on our best configuration. Direct comparison with Sukhbaatar et al. 2018b requires to disable many other components described in this paper **e.g. ABC and multi-goal game structure**. As we already showed, ABC is critical in our setup. Therefore, we can conclude that direct adoption of Sukhbaatar et al. 2018b in our setup will fail drastically as well. In addition, Sukhbaatar et al. 2018b does not study a setup where a policy is trained 100% by asymmetric self-play.
> >
> > **4. Missing pseudocode**
> > Thank you for the nice suggestion! Based on the feedback, we have added pseudocode for asymmetric self-play training in Appendix A.4.
> >
> > **5. Assumption on resettable environment**
> > We are aware of the fact that our asymmetric self-play approach is built on an assumption of using a resettable simulation environment, because Bob needs to start from the same initial state as Alice’s. However we believe this assumption would not be an obstacle to make progress toward training a robot that operates in the real world. Many recent works on sim-to-real transfer ([Sadeghi & Levine, 2017](https://arxiv.org/abs/1611.04201); [Tobin et al., 2017](https://arxiv.org/abs/1703.06907); [OpenAI et al., 2018](https://arxiv.org/abs/1808.00177), [James et al., 2019](https://arxiv.org/abs/1812.07252)) have demonstrated that a robot purely trained in simulation can generalize to the real world. We believe that our approach can be used to train a policy for the real world via similar sim-to-real transfer techniques. We updated Sec. 6 to include a discussion on resettable environments and sim-to-real transfer.
> >
> > **6. Using state observations for experiments**
> > We believe that using state observations does not necessarily make our environment a “toy simulation”. With a potential future research on sim-to-real in mind, we carefully tuned our simulation environment, which will be released together with the paper, to resemble the same robotic setup in the real world. Although we provide accurate state observations for the objects, the policy still needs to learn how to control the robot in an environment with realistic dynamics.
> >
> > Moreover, the main focus of our work is to **explore the zero-shot generalization capability of asymmetric self-play**, and using state observation in experiment does not invalidate our study on the capability of asymmetric self-play approach. We agree that showing the same capability with pure vision policy would consolidate our result even further, but unfortunately we didn’t have enough bandwidth for conducting this additional work.
> >
> > As a final remark, nothing in our framework hinders applying our method to pure vision policy. Although we didn’t use vision policy in this paper, we believe our work is a stepping stone for potential future work that may show a strong zero-shot generalization of a pure vision policy.
> >
> > We hope it can resolve the majority of your confusion. Thank you for reading this far!

---

### Official Review · AnonReviewer1 · 2020-10-28
**Interesting application of self-play for robot manipulation**

**Rating:** 7
**Confidence:** 2

**Review:**

This paper presents a self-play approach to learn a goal-conditioned policy for robot manipulation. The Alice and Bob approach is able to generalise to unseen objects and results in a natural curriculum.
The paper is well written and well structured, the proposed approach is framed within the relevant literature and claims are supported by experimental results.
Experimental evaluation shows that the proposed approach outperforms other methods on several manipulation task. Ablation studies complete the analysis of the proposed approach and help understanding the contribution of each part of the approach.

Comments:
- How does the proposed method compare with other goal-based approaches (e.g. multi-task approaches or goal-based intrinsic motivation exploration)?
- What is the main cause of the low success rate with increasing number of objects? What is the main limitation in the generalisation capability of the approach (Fig. 8)?
- How much does memory (used in policy networks) impact the learned behaviours (especially with several objects to manipulate)?
- Can you discuss the main limitations of your approach with respect to other methods?
- What is the expected transferability of the proposed approach to a real world scenario (e.g. with a real manipulator and real objects)?
- With a self-play approach the exploration is "limited" to what the agent can generate during self-play. How does this affect generalisation in your specific case study?
- Videos are interesting and helpful

---

> ### Author Response · Authors · 2020-11-13
> **Thank you for your reviews and we have responded to each comment below.**
>
> Thank you for your reviews! Based on the feedback, in a new version of the manuscript, we added a discussion on limitations of our work in Sec. 6, mainly on the dependency on a simulated environment and a sim-to-real approach to deploy it to physical robots as future work.
>
> **Regarding the comments:**
> 1. Existing multi-task learning methods assume we have access to a distribution of tasks during training. Our work focuses on zero-shot generalization, which means, we do not program any evaluation task into the training data but rather use self-play to generate a distribution of goals.
> 2. With more objects, the policy has to track the state of more objects, resulting in higher pressure on the policy’s memory. It also demands a more delicate manipulation skills (e.g. moving one object while avoiding touching another) and a sense of ordering (e.g. stacking or rainbow task has to be completed in a certain order).
> 3. It has a positive impact on the generalization performance. When we reduce the size of the LSTM hidden state too much, Bob cannot achieve as good success rates as reported.
> 4. One limitation of our asymmetric self-play approach is that it depends on a resettable simulation environment as Bob needs to start from the same initial state as Alice's. In order to run the goal-solving policy on physical robots, we need to train Alice and Bob in a simulator and take sim2real approach. We have added this into the manuscript.
> 5. Sim-to-real is a natural next step for our work. To deploy the goal solving policy to physical robots, we need to train a vision-only policy and then apply fine-tuning on Bob possibly from domain randomization in sim or data from the physical robots.
> 6. When a task completion requires a delicate sequence of actions, such as using a screwdriver to attach one piece of wood to another, it would become challenging for Alice to discover such a complex goal. Holdout tasks in our experiments are still limited to rearranging or stacking objects. How to discover goals of high complexity is an open question for our future work.
> 7. Thanks! We are happy to hear that.

---

### Official Review · AnonReviewer2 · 2020-10-29
**Goal setting and problem setting agents are brought together to achieve zero-shot generalisation**

**Rating:** 7
**Confidence:** 4

**Review:**

This paper presents an approach to learn goal conditioned policies by relying on self-play which sets the goals and discovers a curriculum of tasks for learning. Alice and Bob are the agents. Alice's task is to set a goal by following a number of steps in the environment and she is rewarded when the goal is too challenging for Bob to solve. Bob's task is to solve the task by trying to reproduce the end state of Alice's demonstration. As a result, the learned policy performs various tasks and can work in zero-shot settings.

Strong points:
- This paper proposes a novel way to generate the goal distribution for training on multiple goals that is rich and scalable. The authors describe the properties of the goals as being achievable and not labor intensive.
- The experimental results are very encouraging.
- The analysis of the method helps to understand which components are important.
- The evaluation on the hold-out tasks is very impressive and pushes the state of the art.
- The paper is well written and very easy to follow, the illustrations are informative and appealing.
- Although this approach is based on previous work on asymmetric self-play, the authors clearly describe the contributions of this work (training clearly from self play, zero-shot generalisation).

Weak points:
- Alice and Bob agents need to perform the task from the same initial state. This might pose some problems when training with a real (not simulated) robot environment.
- The way how goal-conditioning reward is assigned assumes that what matters is the absolute positions of all the objects in the environment. This means that such relative concepts like, for example, "objects being close to each other", cannot be handled in the current framework.
- Alice Behavioural Cloning is currently incorporated into learning by combining the RL and BC losses. Another common way to incorporate demonstrations is to include them in the replay buffer for training a policy. Would it be possible in this method? As this component is crucial for the success of the method, it might be worth investigating the alternatives.
- The baselines of using generative modelling for goal generation are mentioned, but there is no comparison with them. Would it be possible to apply those methods to the studied environments and settings?
- It seems that every policy includes the state as part of its observation. What does this state include? Why can't the policy be trained directly from vision? Does including the state mean that the learning procedure is not applicable to real environments?

I am leaning toward the acceptance of this paper because I find 1) the method interesting and well motivated, 2) experimental results very encouraging and zero-shot generalisation quite impressive. While I still have some concerns (learning from states, starting from the same initial condition), I believe that this paper advances the knowledge and would be beneficial for the research community.

Questions:
- "Multi-goal" game: the experiments show that it improves the results, but I still cannot understand why it is crucial. Does it help because of several goals or because of longer episodes? Currently, it is repeated 5 times, what happens if it is less/more and each game is shorter/longer?
- What happens with multi-goal during testing? Are the tasks split into stages, or the agent solves only for one goal?
- I am a bit confused by step 3 where Bob's reward is computed. It sounds to me now that if Bob fails in the first game, it fails in the whole episode, is it so?
- The policies are not trained directly on the hold-out tasks, but I am wondering, how many such goals (for example, for stacking tasks) appear by accident when Alice sets the goal.
- What about the data complexity of the proposed approach? How many goals did Alice generate, how many steps in each?
- In Fig. 6, why are the results different with a single block and two blocks?


Additional comments:
- I am a bit confused about the content of footnote 1.
- I couldn't see some of the strategies in Fig.4, is there a way to make them more visible?

===Post rebuttal===

I would like to thank the authors for the detailed response and clarification of my questions. I believe that this paper will be valuable for the ML community.

---

> ### Author Response · Authors · 2020-11-13
> **Thank you for the detailed review! We have reply to each points below.**
>
> Thank you for your reviews and questions! Based on the feedback, in a new version of the manuscript:
> 1. We added more clarification on how the goal-conditioned reward is calculated in the main text.
> 2. We added a discussion on the limitations of our work in Sec. 6, mainly on how to make it work on physical robots.
> 3. We added additional comments in Fig 4 to point out that some curves are occluded.
>
> **Regarding weak points:**
> 1. **Sim2real.** We intend to do sim2real transfer in our setting in the future work, potentially with fine-tuning Bob with domain randomization or data collected on the physical robots using the goals discovered during self-play. That’s why we don’t have too much concern on having a resettable simulation environment for training.
> 2. **Absolute state.** We agree that our formulation of goal-conditioned policy focuses on a class of manipulation problems specified by absolute positions and orientations, and cannot handle more general cases such as the example of "objects being close to each other". However, we believe that the capability of solving any goal that can be specified by absolute positions and orientations already covers a wide range of interesting manipulation problems. And it is still an ambitious research objective that the field is actively studying (e.g. [Florensa et al. 2018](https://arxiv.org/abs/1705.06366), [Lynch et al. 2020](https://arxiv.org/abs/1903.01973)).
> 3. **Demo for RL.** We have tried a variant of ABC experiment, in which demonstration trajectories are incorporated into the replay buffer for optimizing PPO loss with an off-policy correction term similar to vtrace (see [IMPALA](https://arxiv.org/abs/1802.01561)). However, this variant didn’t learn anything and its performance was similar to “no ABC” baseline in our paper.
> 4. **Generative modeling.** Unfortunately, applying previous generative modeling based approaches to our environment requires non trivial amounts of extra work. For example, our goal state has a variable length (because of the variable number of objects), and no previous work has used variable length vectors. In addition, it would be impossible to use ABC for Bob if Alice’s goal is produced by a generative model and we have learned that ABC is very crucial for Bob to learn well.
> 5. **Observation space.** The observation space includes robot arm position, gripper position, object state, and goal state. More details on the policy’s observation are listed in Appendix B.3.
> In this work, we intend to explore the zero-shot generalization capability via asymmetric self-play in simulation. Incorporating state into the observation enables both Alice and Bob to learn the ground truth state of the environment and drastically speed up policy training, allowing us to do faster iteration during experimentation. However, nothing in our framework hinders applying our method to pure vision policy. As we are interested in sim2real transfer using the framework proposed in this work, we would like to train a pure vision policy in our framework as future work.
>
> **Regarding questions:**
> 1. **Multi-goal.** We believe multi-goal can encourage better adaptation within one episode. The environmental information that Bob obtains earlier on within the first goal trial, such as the robot dynamics and the number of objects, can help Bob to solve its remaining goals. The effect is also reported in [OpenAI et al. 2019](https://openai.com/blog/solving-rubiks-cube/). Please see more discussion in Appendix A.3.
> 2. **Multi-goal at test time.** At test time, we have 5 random goals per episode for Bob in all the block experiments. We have only 1 goal per episode in several fixed goal holdout tasks such as table setting and rainbow. Details are available in Appendix B.6.
> 3. Yes. If Bob fails in the first game, it fails in the whole episode.
> 4. **Goal complexity.** Stacking happens, as illustrated in Fig. 5, but we don’t have a quantitative metric on how many holdout task goals have been discovered during training yet. In addition, holdout tasks may contain objects that do not appear in the training environment (i.e. see our big hybrid policy experiment), so the exact same goal cannot be found by definition. Qualitatively, Alice seems to find goals that encourage Bob to acquire manipulation skills needed to solve holdout taks (e.g. pushing, grasping, standing up objects, ...).
> 5. **Alice’s goals.** We allow Alice to generate 5 goals per episode unless Alice proposes an invalid goal in the middle (e.g. push the block off the table and then the episode terminates immediately). Alice runs for a fixed number of steps per goal, 100 for state policy and 250 for hybrid policy, reported in Appendix Table 3.
> 6. **1 block vs 2 blocks.** The goal complexity is limited with a single block. With 2 blocks, Alice could propose complex goals like in Fig. 5, which challenges old versions of Bob.

---

> > ### Author Response · Authors · 2020-11-13
> > **(cont'd) Regarding additional comments**
> >
> > 1. We would like to mention the difference between our work and the original asymmetric self-play paper. They use asymmetric self-play as additional data generation steps but the majority of training data still comes from the same task distribution for evaluation.
> > 2. We clarified in the caption that the invisible success rate curves are the ones that achieve zero success rates, and they are occluded by others.

---

### Official Review · AnonReviewer5 · 2020-11-08
**Review for paper "Asymmetric self-play for automatic goal discovery in robotic manipulation"**

**Rating:** 6
**Confidence:** 4

**Review:**

Summary:
This paper uses asymmetric self-play to train a goal-conditioned policy for robot manipulation tasks, which can also generate curriculum automatically and generalize to unseen goals and objects. The experiments contain various challenging manipulation tasks, where the proposed method outperforms all the manually designed curriculum baselines.

Pros:
+ The paper is well-written and easy to follow.
+ The goals and skills discovered by Alice interesting. The proposed method can generate some meaningful goals which work better than explicit curricula.
+ The novel shape/geometry manipulation results are promising.

Questions & Concerns:
- Lack of important baselines. The author uses PPO, but this framework will still support off-policy learning. This adversarial training idea is very similar to CER which is not cited.
  1. (HER) Hindsight Experience Replay
  2. (CER) Competitive Experience Replay (not cited): [1]
- This work is also related to HRL methods, so it needs to include more HRL works in section 4, e.g. HAC[2], HIRO[3]
- (minor) In the "Flip" environment, success is also related to the orientation of the box. What's your rotation representation and the corresponding distance to measure "success"?
- (minor) Are there any exploration strategies in Alice?

In general, this paper proposes to use asymmetric self-play to generate a curriculum for the agent, which is better than the explicit one. However, since there is no comparison with HER-based methods (or HRL), it's hard to justify whether asymmetric self-play is necessary for these kinds of problems.

[1] Liu, Hao, et al. "Competitive experience replay." ICLR 2019

[2] Levy, Andrew, et al. "Learning multi-level hierarchies with hindsight." ICLR 2019

[3] Nachum, Ofir, et al. "Data-efficient hierarchical reinforcement learning." Advances in Neural Information Processing Systems. 2018.

======== POST REBUTTAL RESPONSE========

After reading the feedback and revision of the paper, most of my above concerns are addressed. I agree that the comparison with HER/CER is not fair. I also notice that the author added more references and details based on all 4 reviews.  Thus, I decided to improve my score on this paper.

---

> ### Author Response · Authors · 2020-11-13
> **Thank you for the review! Response to questions & concerns, as well as additional ref.**
>
> Thank you for taking the time to review our paper and helpful suggestions! Based on the review, we have made the following changes in a new version of the manuscript:
> 1. We added missing references on CER, HAC and HIRO, as well as more discussion on the difference between them and our work.
> 2. We added clarification on rotational distance measures.
>
> Response to the reviewer's questions & concerns is detailed below.
>
> **1. Difference with automatic curriculum works.**
> The focus of our work is to achieve zero-shot generalization to many holdout tasks without using them for training. Asymmetric self-play achieves this goal by learning to generate goals with Alice, no pre-defined goal distribution is not necessary for training. On the contrary, HER and CER are designed to use goals sampled from a pre-defined goal distribution, and they have to be trained on a mixture of holdout tasks. Thus, it is not possible to evaluate them without using holdout tasks for training in a zero-shot fashion. Comparing self-play without no prior on task distribution and HER (or CER) trained on a mixture of holdout tasks would not be a fair comparison.
> The fact that we don’t need to design any heuristic based curriculum to make it possible for Bob to learn is a nice “side effect” rather than the main focus of our work. Our main focus is to show that asymmetric self-play can automatically discover goals to zero-shot generalize to unseen holdout tasks.
>
> **2. Additional references.**
> We appreciate the suggestion of additional references. We agree that CER is indeed very relevant to our work because they jointly train two agents to get a goal conditioned policy, while they focus on learning a pre-defined goal distribution. The main focus of CER is to encourage efficient learning and exploration in environments with sparse reward. It provides a boost to some extent when combined with HER, but cannot outperform HER when used alone.
> We also agree that HAC [2] and HIRO [3] are relevant because they use high level policy to generate goals for the low level policy. However, they train the high level policies to maximize the reward for a pre-defined task. Alice in our work generates a goal for Bob without optimizing any predefined task reward.
> We have *added all three suggested citations* and proper discussion into our related work section.
>
> **3. Measure of success for flip task**
> Appendix A.2 describes the success metric in our environment. The orientation of an object is represented by the euler angles of roll, pitch, and yaw for the block. We consider a goal is achieved if the euclidean distance between object and the goal is smaller than 0.04 meters and the Euler angles between object and the goal is smaller than 0.2 radians.
> We originally kept this level of details in the appendix due to the length limit of the main text. In the new version, we added clarification on the rotation measure in the main text, the first paragraph of Sec. 3.1.
>
> **4. Exploration strategies for Alice**
> Exploration of Alice is driven by random action sampling and rewards from asymmetric self-play. Alice is encouraged to discover new and challenging goals for Bob, as Bob can learn to master a goal as training progresses via RL and ABC (Alice behavioral cloning). We do not use any additional exploration strategies for Alice.

---

### Decision · Program_Chairs · 2021-01-07
**Final Decision**

**Decision:**

Reject

**Comment:**

The paper extends previous work on asymmetric self-play by introducing a novel behavior cloning loss (referred to as ABC). The zero-shot results are impressive and demonstrate that the proposed curriculum learning approach pushes the state-of-the-art. The reviewers acknowledge these contributions. The pros of the paper are well summarized by R2,

- The experimental results are very encouraging.
- The analysis of the method helps to understand which components are important.
- The evaluation on the hold-out tasks is very impressive and pushes the state of the art.
- The paper is well written and very easy to follow, the illustrations are informative and appealing.
- Although this approach is based on previous work on asymmetric self-play, the authors clearly describe the contributions of this work (training clearly from self-play, zero-shot generalization).

R1, R2, R5 recommend accepting the paper with scores of 7, 7, 6. R1 expressed that he is not confident about the paper. R4 recommends acceptance with a score of 6. However, R4 also expresses the concern for real-world applicability, "the sim-real gap with respect to applicability to robotics is still a major concern IMO. I have updated my score accordingly." The sim-to-real gap is a concern due to knowledge of perfect state information and the assumption of resets.

Based on confident reviews of R2, R4, and R5, and the impressive zero-shot results, ordinarily, I would recommend the paper to be accepted. However, unfortunately, both the authors and reviewers missed a comparison to prior work, which I detail below. While the current paper makes a good case for zero-shot generalization, it does not compare to previous approaches that also exhibits zero-shot generalization. For instance, Li et al. ICRA 2020 (https://arxiv.org/abs/1912.11032) show that using a simple curriculum that depends on the number of manipulated objects + Graph Neural Networks can generalize very well to unseen tasks. E.g., the results reported in the paper demonstrate that a policy trained on 2/3 blocks generalizes and can stack many more blocks. Their policy learns to stack 6-7 blocks, whereas the paper under review can only stack up to 3 blocks (Figure 8).

The authors mention in Section 5.2 of their paper, "The curriculum:distribution baseline, which slowly increases the proportion of pick-and-place and stacking goals in the training distribution, fails to acquire any skill. The curriculum:full baseline incorporates all hand-designed curricula yet still cannot learn how to pick up or stack blocks. We have spent a decent amount of time iterating and improving these baselines but found it especially difficult to develop a scheme good enough to compete with asymmetric self-play."
This is at odds with results in Li et al.

This reason for dissonance is that good generalization can be achieved by improving two separate components -- the neural network architecture or the learning curriculum. This paper shows good generalization with weak neural net architectures + a good curriculum learning method. It is unclear to me how critical the self-play method would be with a stronger architecture such as a graph network which is arguable more appropriate for the set of tasks presented in the paper. I would like to see if the curriculum is necessary (i.e., complements a stronger architecture) or is it just a replacement for alternate neural network architecture. Without such a study, this paper should not be accepted, because it will add to more noise rather than advancing the field of robotic manipulation.